# A Comparison of Two Hormonal Fertility Monitoring Systems for Ovulation Detection: A Pilot Study

**DOI:** 10.3390/medicina59020400

**Published:** 2023-02-18

**Authors:** Qiyan Mu, Richard Jerome Fehring

**Affiliations:** Institute for Natural Family Planning, College of Nursing, Marquette University, Milwaukee, WI 53233, USA

**Keywords:** mobile applications, fertility monitoring, luteinizing hormone, ovulation detection

## Abstract

*Background and Objectives*: Accuracy in detecting ovulation and estimating the fertile window in the menstrual cycle is essential for women to avoid or achieve pregnancy. There has been a rapid growth in fertility apps and home ovulation testing kits in recent years. Nevertheless, there lacks information on how well these apps perform in helping users understand their fertility in the menstrual cycle. This pilot study aimed to evaluate and compare the beginning, peak, and length of the fertile window as determined by a new luteinizing hormone (LH) fertility tracking app with the Clearblue Fertility Monitor (CBFM). *Materials and Methods*: A total of 30 women were randomized into either a quantitative Premom or a qualitative Easy@Home (EAH) LH testing system. The results of the two testing systems were compared with the results from the CBFM over three menstrual cycles of use. Potential LH levels for estimating the beginning of the fertile window were calculated along with user acceptability and satisfaction. *Results*: The estimates of peak fertility by the Premom and EAH LH testing were highly correlated with the CBFM peak results (R = 0.99, *p* < 0.001). The participants had higher satisfaction and ease-of-use ratings with the CBFM compared to the Premom and EAH LH testing systems. LH 95% confidence levels for estimating the beginning of the fertile window were provided for both the Premom and EAH LH testing results. *Conclusions*: Our pilot study findings suggest that the Premom and EAH LH fertility testing app can accurately detect impending ovulation for women and are easy to use at home. However, successful utilization of these low-cost LH testing tools and apps for fertility self-monitoring and family planning needs further evaluation with a large and more diverse population.

## 1. Introduction

The effectiveness of natural family planning (NFP) methods for avoiding or achieving pregnancy depends on the accurate estimation of the woman’s fertile window during the menstrual cycle. The actual biological fertile window is six days, the day of ovulation and the five days before ovulation [1]. The rise in the estrogen hormone from a developing follicle and the luteinizing hormone (LH) surge are considered the best hormonal markers of the biological fertile window [2]. Advances in urinary hormonal testing have provided women with the ability to objectively and accurately self-estimate their fertile window by home measurement of urinary metabolites of reproductive hormones.

The Clearblue Fertility Monitor ([CBFM], Swiss Precision Diagnostics, GmbH, Geneva, Switzerland) is a hand-held electronic hormonal fertility monitor that estimates a woman’s fertile window by identifying a threshold level of the urine metabolites of estrone-3-glucuronide (E3G) and LH and provides the user with a reading of “low,” “high,” and “peak” fertility. The accuracy of the CBFM in estimating the fertile window of the menstrual cycle has been established by comparison with serial ultrasound of the developing follicle [3,4]. The CBFM has been successfully incorporated into a modern NFP method to effectively help women avoid and achieve pregnancy [5,6,7]. However, the CBFM and monthly test strips it uses can be rather expensive and the monitor only provides qualitative levels of the hormones. 

Researchers have shown that inexpensive urinary LH test kits among women using NFP enhanced the ease and objectivity of determining the peak of fertility and the beginning of the infertile phase of the menstrual cycle [8]. Other researchers determined that women’s self-rating LH test strips with results as low, high, and peak fertility correlated closely with the LH peak of the CBFM and provided a more accurate indication for the beginning of the fertile window of the menstrual cycle than the CBFM [9]. However, some participants had difficulty in estimating LH test results by visual observation of the color of the lines. 

The Premom Ovulation Tracker app (Easy Healthcare Corp, Chicago, IL, USA) measures the urinary LH hormone with LH test strips using a cell phone camera and synchronized app that visually displays the hormonal levels throughout a menstrual cycle. The Premom app offers two types of urine LH test strips. One is the Premom LH test strip which provides a quantitative level of the LH in the urine. The second is the Easy@Home (EAH) LH test strip which provides a ratio level of the LH in the urine. Combining urinary hormone tests with the convenience of a mobile app could be beneficial for women who seek to monitor their fertility. However, the Premom Ovulation Tracker app has not been evaluated for accuracy in detecting ovulation and estimating the fertile window of the menstrual cycle by independent researchers. Furthermore, the Premom Ovulation Tracker app has not been tested for use in NFP, specifically for helping women and couples avoid pregnancy. Therefore, the primary objective of this study was to compare the beginning, peak, and length of the fertile window as determined by the CBFM and the Premom Ovulation Tracker app. The second objective was to compare the satisfaction and ease of use with the CBFM and the Premom Ovulation Tracker app. 

## 2. Materials and Methods

This pilot study used a prospective descriptive design. Institutional review board approval was obtained from the University. Study inclusion criteria for women are (1) between the ages of 18 and 42, (2) menstrual cycle length of 21 to 42 days, (3) had not used depot medroxyprogesterone acetate over the past 12 months, (4) had no history of oral or subdermal contraceptives for the past 3 months, (5) had at least three cycles past breastfeeding weaning, and (6) have no known fertility problems.

The study participants were recruited through email using the Marquette NFP teacher network throughout North America. The teachers shared the study information with their clients, and the interested clients contacted the researchers. Women were pre-screened with questions based on the inclusion criteria using a simple survey. Thirty participants were recruited by this snowball method. After receiving informed consent, the participant was assigned a study ID and then randomized 1:1 into either the Premom LH test group or the EAH LH test group. 

All study participants were currently using the CBFM to track their ovulation to avoid pregnancy. Each Premom LH test group participant was provided with 60 Premom LH test strips and each EAH LH test group participant was provided 60 EAH LH test strips. Information was provided to the participants on how to download and use the Premom Ovulation Tracker app and urine testing. 

Participants were instructed to start testing their first morning urine void on day six of their menstrual cycle. They continued morning urine testing for 20 days with each menstrual cycle and for a total of three cycles. On the test morning, urine was collected in a clean container and the test strips from each monitoring system were placed in the urine for 15s consecutively (i.e., first the CBFM test strip and then right after the Premom or EAH test strip based on the group allocation). The participants recorded the low, high, and peak results for the CFBM and the quantitative results for the Premom LH test strip or the ratio result from the EAH LH test strip on their charting sheet.

At the end of the third menstrual cycle, the participants completed a survey to evaluate user acceptability and satisfaction using the electronic fertility monitor. The survey was developed by Severy et al. [10,11,12]. The survey contained eight questions that assessed whether a fertility monitoring system is acceptable, easy to use, non-invasive, and convenient for in-home use and provides clear and objective results. The survey items were ranked on a scale from 1 to 7, with a higher number indicating greater ease of use and satisfaction with the fertility monitor.

Descriptive statistics were calculated for participant demographics and the characteristics of the menstrual cycles. Independent t-tests were used to determine if there were differences between the Premom LH and EAH LH groups. Pearson correlation analysis was used to assess the correlation of the Premom peak LH level with the first CBFM peak day and the correlation of the EAH peak LH level with the first CBFM peak day. Paired t-tests were used to compare differences in ease of use and satisfaction for the CBFM, the Premom LH, or the EAH LH testing. *p* ≤ 0.05 was set as the a priori significance level for the overall test. A Bonferroni correction (*p* ≤ 0.006) was used to control the increased error rate for comparing multiple question items on the satisfaction survey. All statistical analyses were conducted using IBM Statistical Package for Social Science software (SPSS Version 28.0, IBM Corp, Armonk, NY, USA).

## 3. Results

### 3.1. Participant Demographics

A total of 30 women consented to participate. The mean age was 33.47 (range 26–42, SD 4.76). All participants were Caucasian and had some college education. All 30 participants initiated the testing and charting. One participant achieved a desired pregnancy after the first cycle and withdrew from the study. The participant did complete the satisfaction survey and agreed that her first cycle data and survey data could be used for the study. The demographics of the 30 participants can be found in Table 1. There were no statistical differences in the demographic variables between the Premom LH Group and the EAH LH group.

### 3.2. Cycle Data and Ovulation Detection

The 30 participants contributed 84 menstrual cycles. The average cycle length was 28.82 days (range 24–38, SD 3.62). The LH surge was detected in 94% of the cycles by the CBFM, 82% by the Premom LH test strips, and 95% by EAH LH test strips. Table 2 shows the menstrual cycle parameters. There were no differences in the estimated day of ovulation (as indicated by the CBFM) between the two study groups.

The correlation of the Premom peak day with the CBFM first peak was r = 0.99; *p* < 0.0001, and the correlation of EAH LH peak with the CBFM first peak was also r = 0.99; *p* < 0.0001. The fertile window was estimated for each cycle based on the high and peak reading of the monitors and a count of three days past the peak LH Day. There were significant differences in the estimated length of the fertile window when comparing the Premom LH testing result with the CBFM and the EAH result with the CBFM for both groups. For the Premom LH group, the estimated Fertile window based on the Premom LH was 2.36 days (SD 0.88) and the CBFM was 8.19 days (SD 3.93). For the EAH group, the estimated fertile window based on the EAH LH was 2.53 days (SD 0.86) and the CBFM was 7.53 days (SD 3.60).

The estimated day of ovulation for the CBFM was defined as the second peak day on the CBFM. Table 3 and Table 4 displayed the mean scores and 95% confidence interval (CI) of the Premom LH and the EAH LH levels five days before the second CBFM peak day and three days after. The Premom LH and the EAH LH testing had the highest scores on the CBFM’s first peak day, which was the day that the CBFM picked up a threshold level of LH. The threshold level of LH based on the 95% CI five days before the estimated day of ovulation was 2.24 for the Premom quantitative result and 0.17 for the EAH LH ratio level result.

### 3.3. User Satisfaction and Ease-of-Use Rating

Overall, the participants reported significantly higher satisfaction and ease-of-use ratings for the CBFM compared to the Premom LH testing or the EAH LH testing. For the Premom LH group, the mean score for the CBFM was 51.29 (SD 4.56) vs. 43.07 (SD 11.82) for the Premom testing (t = 2.66; df = 13; *p* < 0.02). For the EAH group, the mean score for the CBFM was 52.38 (SD 4.45) vs. 44.67 (SD 9.05) for the EAH testing (t = 2.79; df = 11; *p* < 0.02).

When comparing the individual satisfaction and ease-of-use items, the only items that were significantly different for the Premom LH group were Item 5, “To what extent, if at all, do you think using the NFP fertility monitoring system has increased your ability to avoid pregnancy?” and Item 8, “At this point in time, how do you like the fertility monitoring system?” (Table 5). The only significant item for the EAH group was Item 4, “What would you say is your overall opinion of the fertility monitoring system?” (Table 6).

Qualitative feedback from the participants showed mixed feelings toward the Premom Ovulation Tracker app and its LH testing kits. One study participant shared, “I really like the premom numeric rating given to the picture of the test, it makes interpretation so much easier. My only frustration was that sometimes it was difficult to get a good picture due to an old phone and poor light.” Another participant stated, “Overall, I found the system easy to use and read, but I dislike having to time the test myself. The Clearblue monitor is much easier to use because I can walk away while the test is reading. Much easier for busy mornings.”

## 4. Discussion

In the 1980s, scientists determined that one of the best predictors of the beginning of the fertile phase was the rise in estrogen and the urine metabolite E3G [13]. The CBFM is based on technology that uses immunoassay techniques to measure the rise in E3G to predict the beginning of the fertile phase. However, these monitors can be expensive to purchase along with the use of expensive test strips. A need for cheaper but accurate measures of the fertile phase would be welcome. Our findings show the correlation of the CBFM peak in LH with the Premom LH, and the EAH LH peak was excellent at r = 0.99, *p* < 0.001. However, the first high readings of the Premom and EAH did not correlate well with the CBFM first high reading. That is reflected in the significant difference in the total days of high and peak fertility provided by the two monitoring systems, with the Premom LH testing and EAH LH testing providing about two days of peak and high fertility, and the CBFM about five days. One explanation for the significant difference in the estimated fertile window is that the Premom LH testing and the EAH LH testing estimates of the fertile window were based on the measure of LH and the CBFM on the rise in estrogen (E3G).

Both monitoring systems are intended for women who want to become pregnant and target the most fertile days of the menstrual cycle. Premom and EAH LH testing have a tighter estimate of two days. The Premom and EAH LH peak days correlate strongly with the peak days of the CBFM. These two days have the greatest probability of fertility during the six-day fertile window [1,14,15]. The information is sufficient for women who want to time their intercourse for conception. However, this short fertile notification is not enough for women who wish to avoid pregnancy, instead they would need to combine these findings with the Marquette Method algorithm to determine the beginning of the fertile window. Following the previously published estimate rule for the beginning of the fertile window (i.e., the day of ovulation and the five days preceding ovulation), we calculated the mean LH levels for Premom and EAH for five days before the estimated day of ovulation (i.e., second peak day of the CBFM). These two LH levels could be used to estimate the beginning of the fertile window for Premom users and EAH users. However, given the small sample size, these numbers should be validated with further research using a larger sample.

Although we hypothesized that there would be no differences in the satisfaction and ease of use of the compared fertility monitoring systems, the two LH monitoring systems (i.e., Premom and EAH) had significantly lower scores. The lower scores make sense since all the participants had been using the CBFM for family planning purposes and were familiar with its use. The CBFMs were their personal use monitors, whereas the Premom and EAH LH testing were new to them. A fairer comparison of the CBFM with the Premom and EAH for ease of use and satisfaction would be to randomize women users who have not used either system into a CBFM, Premom, and EAH group. Since we only used current CBFM users, there most likely was an obvious bias to that system.

More women are interested in utilizing mobile health tools and apps in their fertility and reproductive health self-care [16,17]. Combining LH testing with a synchronized app to track a woman’s fertility offers potential advantages in cost and time saving for women who wish to use these tools over a long period of time. However, there are concerns about the convenience of testing and accuracy of using the camera on the phone for the LH reading for the Premom Ovulation Tracker app. Future study needs to evaluate these issues with a large and diverse woman population.

## 5. Conclusions

There is a boom in reproductive mobile health monitoring systems and more women are exploring these options for fertility monitoring or family planning purposes [18,19]. The use of mobile applications for fertility tracking requires scrutiny and evaluation [20,21]. This study only focused on the accuracy and user experience of the Premom Ovulation Tracker app and the two types of LH testing strips. We did not evaluate the comprehensiveness and other aspects of the app [19,21]. These low-cost LH testing tools and apps can be successfully incorporated into a woman’s fertility self-monitoring with clearly defined fertility algorithm and guidance. Health providers caring for women need to stay up-to-date with current technologies and evidence to provide patient education and counsel for fertility management and family planning.

## Figures and Tables

**Table 1 medicina-59-00400-t001:** Comparison of the demographic parameters between the two study groups.

	Premom LH Group *n* = 15	Easy@Home LH Group *n* = 15	*p* Value
Age	33.87 (5.15)	33.07 (4.50)	0.65
BMI	26.13 (1.31)	23.57 (1.00)	0.13
Years in Marriage	8.07 (5.13)	7.13 (3.20)	0.56
Total Pregnancies	3.07 (2.19)	2.93 (1.94)	0.86
Living Children	2.60 (1.80)	2.40 (1.81)	0.72
Miscarriages	0.47 (0.92)	0.60 (1.30)	0.75

Abbreviations: BMI, body mass index; LH, luteinizing hormone.

**Table 2 medicina-59-00400-t002:** Menstrual cycle parameters for the two study groups based on the CBFM.

	Premom LH Group *n* = 43 Cycles	Easy@Home LH Group *n* = 41 Cycles	*p* Value
Cycle Length	29.35 (3.35)	28.38 (3.91)	0.23
Length of Menses	4.93 (1.18)	4.98 (0.88)	0.84
Length of Follicular Phase	16.62 (3.39)	16.63 (3.11)	0.98
Length of Luteal Phase	12.82 (1.68)	11.84 (1.53)	0.01
Estimated Day of Ovulation	15.62 (3.39)	15.65(3.17)	0.96

Abbreviations: CBFM, Clearblue Fertility Monitor; LH, luteinizing hormone.

**Table 3 medicina-59-00400-t003:** Premom quantitative LH levels in relation to the pre- and post-second CBFM peak day.

Days Relative To 2nd CBFM Peak Day	Premom LH Level	95% CI
Pre-5	2.59 (1.09)	2.24–2.93
Pre-4	2.87 (0.68)	2.67–3.10
Pre-3	3.04 (1.10)	2.69–3.40
Pre-2	4.53 (2.76)	3.73–5.53
Pre-1	19.57 (12.33)	16.15–24.50
2nd CBFM peak day	10.66 (6.31)	8.60–12.70
Post-1	5.13 (3.89)	3.91–6.49
Post-2	3.49 (1.95)	2.86–4.16
Post-3	3.04 (1.81)	2.47–3.64

Abbreviations: CBFM, Clearblue Fertility Monitor. Note: second CBFM peak day is defined as the estimated day of ovulation.

**Table 4 medicina-59-00400-t004:** Easy@Home ratio LH levels in relation to the pre- and post-second CBFM peak day.

Days Relative to 2nd CBFM Peak Day	Easy@Home LH Level	95% CI
Pre-5	0.20 (0.06)	0.17–0.24
Pre-4	0.20 (0.07)	0.17–0.23
Pre-3	0.23 (0.11)	0.19–0.27
Pre-2	0.33 (0.12)	0.28–0.38
Pre-1	1.17 (0.47)	1.00–1.38
2nd CBFM peak day	0.82 (0.32)	0.71–0.94
Post-1	0.48 (0.20)	0.41–0.55
Post-2	0.39 (0.20)	0.31–0.45
Post-3	0.33 (0.14)	0.28–0.37

Abbreviations: CBFM, Clearblue Fertility Monitor. Note: second CBFM peak day is defined as the estimated day of ovulation.

**Table 5 medicina-59-00400-t005:** Ease of use and satisfaction for Premom LH testing versus CBFM.

Satisfaction Items	Premom LH Testing	CBFM Testing	*p* Value
Item 1	5.93 (1.03)	6.47 (0.64)	0.07
Item 2	5.93 (1.22)	6.53 (0.64)	0.08
Item 3	6.07 (1.79)	6.53 (0.74)	0.30
Item 4	5.13 (1.85)	6.40 (0.74)	0.02
Item 5	4.73 (2.13)	6.80 (0.41)	<0.001
Item 6	4.71 (2.13)	5.71 (1.82)	0.18
Item 7	5.93 (1.53)	6.67 (0.62)	0.10
Item 8	4.73 (1.98)	6.33 (0.82)	0.005

Abbreviations: LH, luteinizing hormone; CBFM, Clearblue Fertility Monitor.

**Table 6 medicina-59-00400-t006:** Ease of use and satisfaction for Easy@Home LH monitoring versus CBFM.

Satisfaction Items	Easy@Home LH Testing	CBFM Testing	*p* Value
Item 1	5.47 (1.51)	6.47 (0.64)	0.02
Item 2	5.93 (1.14)	6.50 (0.65)	0.06
Item 3	5.87 (1.36)	6.50 (0.73)	0.02
Item 4	5.53 (1.18)	6.53 (0.64)	0.004
Item 5	5.29 (1.73)	6.71 (0.61)	0.02
Item 6	5.07 (1.86)	6.50 (0.85)	0.02
Item 7	5.80 (1.32)	6.47 (0.64)	0.10
Item 8	5.29 (1.44)	6.35 (0.63)	0.02LH

Abbreviations: LH, luteinizing hormone; CBFM, Clearblue Fertility Monitor.

## Data Availability

Not applicable.

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
