# Peer review of "A Comparison of Two Hormonal Fertility Monitoring Systems for Ovulation Detection: A Pilot Study"

_medicina, 2023, doi:10.3390/medicina59020400_

Round 1

Reviewer 1 Report

This is an interesting study, well presented, and an important addition. 

I did see one correction needed. The data in table 6 do not agree with the text in 171-172; table 6 shows significant p values for items 5&8 but not item #4. In both tables 5&6 the p values are identical which begs the question, if a transcribing error occurred while creating these tables.

If space allows, it would be useful to elaborate on the statements regarding threshold levels that could be used to define the fertile window using Premom LH (2.24) and EAH (0.15).  You have data on several cycle days before "pre-1"; how many of these days were above these putative cutoff's values? Some additional text would be enlightening.

Author Response

Dear Reviewer, 

Thank you for the thoughtful and detailed review of our manuscript. The reviewers provided excellent and constructive feedback. We have addressed the reviewers' and editors' comments as outlined in the portal. The revision is marked with track change. We also attached a detailed response table addressing the reviewers' comments. Please let us know if you have any questions regarding the revision. 

Reviewer 2 Report

“…(2) menstrual cycle length of 21 to 42 days;…” Please clarify how menstrual cycle length was assessed. What question was asked of the participants?

“At the end of the third menstrual cycle, the participants completed a survey developed by Severy…” What is Severy?

There is no reason to include t-tests for randomized group differences at baseline. Consistent with CONSORT guidelines, I recommend removing them: https://www.consort-statement.org/checklists/view/32--consort-2010/510-baseline-data.

What are the units for LH in Tables 3 and 4? Is that an absolute level?

In Tables 3 and 4, is the “CBFM Peak Day” the day of ovulation? Or would ovulation be the day after the peak day? Similarly, would ovulation be estimated to occur 24 hours after the peak Premom LH level? 24 hours after the Easy@home LH level?

Instead of listing the “Satisfaction Items” as “Item 1,” “Item 2,” etc., can you apply brief descriptions of each item in Tables 5 and 6? Right now, they are not very informative. To add room in the tables you could remove the p-values and make them footnotes.

In the results section, the length of the “fertile window” is described, but this was not defined as an endpoint in the Methods. How were the beginning and end of the fertile window identified?

The correlation coefficients are informative, but it would be nice to see some measures of accuracy and agreement such as sensitivity, specificity, positive predictive value, negative predictive value and/or kappa statistics.

Author Response

(The authors gave the same response as above.)
